# Protective Effects of α-Lipoic Acid and Chlorogenic Acid on Cadmium-Induced Liver Injury in Three-Yellow Chickens

**DOI:** 10.3390/ani11061606

**Published:** 2021-05-29

**Authors:** Jiabin Shi, Xiaocui Chang, Hui Zou, Jianhong Gu, Yan Yuan, Xuezhong Liu, Zongping Liu, Jianchun Bian

**Affiliations:** 1College of Veterinary Medicine, Yangzhou University, Yangzhou 225009, China; mx120200948@yzu.edu.cn (J.S.); xiaocui_chang@163.com (X.C.); zouhui@yzu.edu.cn (H.Z.); jhgu@yzu.edu.cn (J.G.); yuanyan@yzu.edu.cn (Y.Y.); Liuxuezhong68@163.com (X.L.); liuzongping@yzu.edu.cn (Z.L.); 2Jiangsu Co-Innovation Center for Prevention and Control of Important Animal Infectious Diseases and Zoonoses, Yangzhou 225009, China; 3Joint International Research Laboratory of Agriculture and Agri-Product Safety of the Ministry of Education of China, Yangzhou University, Yangzhou 225009, China

**Keywords:** chicken liver damage, cadmium, α-lipoic acid, chlorogenic acid, oxidative stress

## Abstract

**Simple Summary:**

Cadmium (Cd) exerts pernicious influences on global health. We evaluated the protective effects of α-lipoic acid (α-LA) or chlorogenic acid (CGA) and their combination on counteracting Cd toxicity in vivo in three-yellow chickens. Administration of Cd (50 mg/L) alone lowered the production performance and resulted in biochemical, histologic and enzyme changes within the liver consistent with hepatic injury induced by oxidative stress and apoptosis of hepatocytes. However, the above variations of the Cd group were partially or fully reversed by administration of either α-LA or CGA; their combination showed an even better effect in attenuating Cd-induced hepatotoxicity. This study provided a practical and feasible approach to rescuing Cd intoxication in animal production.

**Abstract:**

Cadmium (Cd) is a type of noxious heavy metal that is distributed widely. It can severely injure the hepatocytes and cause liver dysfunction by inducing oxidative stress and mitochondrial damage. We evaluated the protective effects of α-lipoic acid (α-LA) or chlorogenic acid (CGA) and their combination on counteracting cadmium toxicity in vivo in three-yellow chickens. For three months, CdCl_2_ (50 mg/L) was administrated through their drinking water, α-LA (400 mg/kg) was added to feed and CGA (45 mg/kg) was employed by gavage. The administration of Cd led to variations in growth performance, biochemical markers (of the liver, kidney and heart), hematological parameters, liver histopathology (which suggested hepatic injury) and ultrastructure of hepatocytes. Some antioxidant enzymes and oxidative stress parameters showed significant differences in the Cd-exposure group when compared with the control group. The groups treated with Cd and administrated α-LA or CGA showed significant amelioration with inhibited mitochondrial pathway-induced apoptosis. Combining both drugs was the most effective in reducing Cd toxicity in the liver. In summary, the results demonstrated that α-LA and CGA may be beneficial in alleviating oxidative stress induced by oxygen free radicals and tissue injury resulting from Cd-triggered hepatotoxicity.

## 1. Introduction

Cadmium (Cd), one of the most common toxic heavy metal pollutants in the environment, can enter an animal’s body and cause serious damage to the liver, kidney, skeleton and other tissues and the immune, reproductive and cardiovascular systems [1]. In chickens, studies have demonstrated that the bioaccumulation of Cd can cause severe damage to multiple organs such as brains, lungs, hearts, livers, kidneys, ovary and testis [2]. Cd accumulates in various tissues and organs because of its prolonged biological half-life and low excretion rate from the body, thereby threatening human health through transmission via the food chain and bioaccumulation [3]. Being an important organ of detoxification in the body, the liver is a target organ of this toxic metal [4]. Increasing evidence has shown that acute exposure to Cd can cause a variety of pathological changes in the liver, including total weight, hepatic enzyme activities, oxidative stress indexes and histological parameters. Histopathological damage in the liver is demonstrated through the disorganization of the hepatic parenchyma, swelling, degeneration and vacuolization of the hepatocytes and focal necrosis [5]. Chronic exposure to Cd can lead to severe adverse effects such as hepatic and renal dysfunctions, osteoporosis and several types of cancer [6]. Studies have shown that both acute and chronic Cd poisoning can cause considerable damage to the liver of animals.

When Cd enters the body (mainly through the digestive tract or lung), it is bound to metallothionein (MT), which contains a sulfhydryl group. As a result, a Cd–MT complex is formed, which is then slowly released back into circulation [7]. In the mitochondria, Cd rapidly accumulates, leading to mitochondrial dysfunction, production of reactive oxygen species (ROS), release of cytochrome c (cyt c), caspase activation, apoptosis and necrosis of the hepatocytes [8]. Studies have shown that oxidative stress is one of the mechanisms responsible for Cd toxicity, which leads to the changes in antioxidant activities, production of lipid peroxide and disruption of intracellular sulfhydryl homeostasis [9]. Some studies indicate that Cd can mimic the functions of some divalent cations, resulting in metabolic abnormalities in the hepatocytes and injuries to the liver [10]. Cd also induces DNA strand breaks, which activates the DNA damage response signaling the ATR-CHK1-p53 pathway [11]. Mitochondria are the target of Cd-induced cytotoxicity [12]. Accumulation of Cd in the body increases mitochondrial permeability and swelling, leading to mitochondrial dysfunction and inhibited respiration [13]. Previous research has demonstrated that the expression of p38 protein increased significantly in cancer cells exposed to Cd [14]. However, the exact mechanism involved remains unclear. Therefore, in this study, we investigated Cd toxicity in the liver of chickens.

Due to the widespread distribution of Cd and the severity of Cd poisoning, there is an increasing interest in antidotes for Cd intoxication therapy. Heavy metal chelating agents are initially used to relieve Cd toxicity, followed by the administration of various vitamins and minerals. Some chelating agents (such as EDTA or BAL) antagonize Cd-induced hemorrhagic necrosis in mice; Zn also appeared to have a protective effect on reducing Cd absorption and accumulation and antagonizing Cd-induced hepatotoxicity in newly hatched chicks [15]. Some studies have demonstrated that blueberry extract (with a powerful antioxidant capacity) can protect the liver by activating liver antioxidative enzymes, lowering glutathione (GSH) concentration and reducing inflammation by decreasing nitric oxide (NO) content [16]. α-lipoic acid (α-LA) and chlorogenic acid (CGA) are two strong antioxidants capable of normalizing the oxidative stress and improving organ functions [17,18].

B vitamins are usually used as antioxidants because of their excellent antioxidant activities. α-LA is a coenzyme that belongs to the vitamin B family that exists in the mitochondria [19]. Some researchers have demonstrated that α-LA (often called a super antioxidant) reduces the amount of free radicals or ROS in the body, for example, NO, hydrogen peroxide and hydroxyl radical [20]. It also aided the regeneration of some endogenous antioxidants such as oxidized ascorbate, GSH, coenzyme Q and vitamin E, which were reproduced after Cd-poisoned male rats were treated with α-LA [21]. The liver is very effective in taking up α-LA, and its metabolites can be discarded from the body through bile [22]. Various studies have proved that a low dose of α-LA is safe and non-toxic, and it is now widely used in animal husbandry and medical treatments. 

CGA, a secondary metabolite of aerobic respiration in plants, is used extensively for its anti-inflammatory, antiviral, antitumor, antibacterial and antioxidant effects [23]. Due to CGA’s functional diversity and the fact that it is one of the main polyphenols in the human diet, it has been widely applied in food production. To date, limited studies have examined the effects of CGA in poultry tissues. Therefore, more studies should be done in the future to access the possibility of CGA usage in Cd-exposed chickens. This kind of phenolic compound has been proven for the first time to improve growth performance of specific-pathogen-free (SPF) chickens and ameliorate blood biochemical indexes [18]. The 1000 mg/kg CGA supplementation increased the activities of superoxide dismutase (SOD) and nuclear factor erythroid-2 related factor 2 (Nrf2, an important transcription factor involved in regulating oxidative stress in cells) and reducing the content of MDA, thus improving oxidative stability in stressed broilers [24]. Similar results were observed in this study.

Recently, an increasing number of theories and studies involving antidotes to Cd have been conducted. Among them, α-LA and CGA stand out due to strong antioxidant capacity without any toxicity under a certain dose. In comparison with mice or rats in many studies relevant to Cd, we select traditional economic animals, chickens, as our experimental model, which are of greater importance in animal production in reality. Our research aims to compare the antagonistic effect of α-LA, CGA and their combination on growth performance, biochemical indexes, histological changes, antioxidative potential and the mitochondrial apoptosis pathway in Cd-treated chickens to assess their capacity to alleviate Cd-induced liver injury. In view of the extensiveness and severity of Cd pollution, this study provides a feasible approach to treating Cd intoxication in animal production. 

## 2. Materials and Methods

### 2.1. Animal Treatments and Chemicals

A total of 96 healthy three-yellow chickens (aged 1 day) were obtained from Jiangsu Provincial Poultry Research Institute (Yangzhou, China). Chickens were acclimatized for one week prior to experimentation (at 30 °C and 70% humidity). Then, the chickens aged 9 days were sorted by body weight (BW). Similar BW chickens were randomly divided into eight groups containing 12 animals (six males and six females): blank control, Cd, α-LA, CGA, α-LA + CGA, Cd + α-LA, Cd + CGA and Cd + α-LA + CGA. Chickens in the same group were reared in three different cages, so each cage got four samples, which were then mixed into one sample. At last, each group got three samples (*n* = 3). The ambient temperature was dropped one degree every three days until it reached 23 °C, while the humidity remained unchanged. Each group was provided sufficient water and food. Each day, CdCl_2_ (50 mg/L) was administrated orally [25], α-LA (400 mg/kg) [26] was added to the feed and CGA was administrated by gavage (45 mg/kg) [27]. The animals were monitored once a day for 90 days for body mass, water and food consumption and clinical manifestations. After the final treatment, they were fasted for 12 h and then euthanized following the protocols and ethical procedures. The study was approved by the Yangzhou University Animal Care and Use Committee (Approval ID: SYXK (Su) 2017–0044). The serum was separated and diluted 10 times with DDW, then the concentration of cadmium was determined by inductively coupled plasma mass spectrometry (ICP–MS). The heart, liver, kidney and brain were dissected out and weighed by electronic balance to calculate organ coefficients. Organ coefficient % = (organ weight/body weight) × 100%.

CdCl_2_·2.5H_2_O and diaminobenzidine (DAB) solution were obtained from Shanghai Macklin Biochemical (Shanghai, China). α-LA and CGA were purchased from Shanghai yuanye Bio-Technology (Shanghai, China). Bax was from GeneTex (Irvine, CA, USA). 

### 2.2. Measurement of Serum Biochemical Indexes 

The main biochemical indexes of organ functions in chicken serum were measured by the AU5800 automatic blood biochemical analyzer (Beckman Coulter, America). The indexes about liver function included total protein (TP), albumin (ALB), globulin (GLB), ratio of ALB and GLB (A/G), aspartate aminotransferase (AST) and alkaline phosphatase (ALP). Biochemical indicators of renal function: creatinine (CREA) and uric acid (UA). Cardiac biochemical markers: creatine creatinine (CK). Blood routine test: red blood cells (RBC) and HGB concentration.

### 2.3. Histopathological Studies 

Specimens were dissected to obtain fresh liver tissue, which was then fixed in formalin-saline (each sample size was 2 × 2 × 0.3 cm³). Samples were then rinsed with running water for 12–24 h and dehydrated with different concentrations of alcohol and xylene. The samples were then embedded in paraffin, sliced and dried in an oven. The samples were then stained with hematoxylin and eosin (H&E) dye and sealed with neutral gum for microscopical observations. The DMI3000B inverted microscope was from Leica, Wetzlar, Germany.

### 2.4. Observation of the Ultrastructure of Liver

First, fresh tissue was taken out and fixed in 2.5% glutaraldehyde at 4 °C overnight (12 h). The tissue was repeatedly rinsed in 0.1 mol/L phosphate buffer saline (PBS) for 15 min three times. The samples were then fixed with 1% osmic acid. Next, 0.1 mol/L PBS was used to rinse the samples for 15 min; rinsing was repeated 3–4 times per sample. Dehydration was carried out with a gradient concentration of ethanol and 100% propanol. The samples were then immersed in 100% acetone, embedded in the resin and dried at 60 °C for 48 h. A slice was then prepared. Lastly, the slice was stained with uranium peroxide and lead citrate prior to observation using HT7800 electron microscopy (Hitachi, Tokyo, Japan).

### 2.5. Measurement of Oxidants and Antioxidants 

First, 80–100 mg of liver tissue was rinsed with 1 mL of normal saline, centrifuged at 2500 r/min for 3 min and the supernatant was discarded. Then 9 mL of normal saline was added and the tissue block was homogenized in a manual homogenizer and then centrifuged. The supernatant was 10% tissue homogenate. Finally, the content of lipid peroxidative product was measured by the malondialdehyde (MDA) assay kit. The enzymatic activities of liver tissue were measured according to the instructions of the total superoxide dismutase (T-SOD) assay kit, catalase (CAT) assay kit, hydrogen peroxide (H_2_O_2_) assay kit, glutathione S–transferase (GSH-ST) assay kit, reduced glutathione (GSH) assay kit, glutathione peroxidase (GSH-Px) assay kit and total antioxidant capacity (T-AOC) assay kit. All kits were purchased from Nanjing Jiancheng Bioengineering Institute (Nanjing, China).

### 2.6. Determination of Trace Elements in Liver

A total of 200 mg of dried tissue was dissected, minced by mortar and transferred to the digestion tube. A total of 4 mL of superior nitric acid was added into it. Then the microwave-assisted digestion program began (Multiwave GO was from Anton Paar, Trading Co., Ltd, Shanghai, China). The digested solution was transferred to an acid-driven processor at 150 °C for 40 min and then to BHW-09A constant temperature digester (Botonyc, Shanghai, China). Then the solution was added with double-distilled water (DDW) to 10 mL for follow-up tests. According to Pinheiro et al. [28] Elan DRC-e ICP-MS (PerkinElmer, Waltham, MA, USA) detection was established for determination of trace elements (Mn, Fe, Cu, Zn and Se) in livers of chickens.

### 2.7. Immunohistochemical Assay of Bax Protein

The paraffin sections prepared in the same way as 2.3 were decolorized first. The sections were incubated with 0.3% H_2_O_2_ deionized water for 30 min to inactivate endogenous peroxidase prior to washing with PBS. Serum was employed for antigen-specific blocking. The Bax antibody was diluted 1:50 before adding it to the slides. The slides were maintained at 4 °C overnight (12 h). Each slide was then washed in PBS for 3 min, five times. The slides were then incubated with a secondary antibody and then washed again. Streptavidin-peroxidase was added to the slides for at least 30 min at 37 °C. DAB solution was used to color the samples. The samples were then washed with tap water, redyed with hematoxylin and sealed in neutral resin. All the slides were then observed under an upright fluorescence microscope (Olympus, Tokyo, Japan).

### 2.8. Total RNA Extraction and qRT-PCR

According to the mRNA sequence of chicken Bax, Bcl-2, Cyt C, Caspase 3, Caspase 9 and β-actin in GenBank, the primers for QRT-PCR were designed by Oligo and synthesized by Beijing Genomics Institution (shown in Table 1). A total of 25–30 mg of liver was used to make tissue homogenate. RNA was extracted by TRIzol (Vazyme Biotech Co, Nanjing, China), and converted to cDNA via a reverse transcription kit HiScript Q RT SuperMix (Yeasen Biotech Co, Shanghai, China) for qPCR. Using the SYBR Premix Ex Taq TM II kit, qRT-PCR was performed with the 7500 Real-Time PCR System (Bio-Rad, Hercules, CA, USA). According to the mathematical model proposed by Pfaffl, the fold change of target gene expression was calculated using the formula: 2^−∆∆Ct^.

### 2.9. Statistical Analysis

All data are presented as the mean ± SD (*n* = 3). Data was analyzed using SPSS 22.0 (Copyright© 2019, IBM Crop., Armonk, NY, USA) statistical software and plotted using GraphPadPrism 6.0 (Copyright© GraphPad Software, Inc., San Diego, CA, USA). The significant differences between the different groups were compared using a one-way ANOVA. A result of *p* < 0.05 was considered a significant difference; *p* < 0.01 indicated an extremely significant difference.

## 3. Results

### 3.1. Effects of α-Lipoic Acid and Chlorogenic Acid on the Growth Performance of Cadmium-Poisoned Chickens 

The effects of α-LA and CGA on the BW of Cd-poisoned chickens are provided in Table 2. At each stage of the experiment (30, 60 and 90 days), compared with the blank control group, the body mass of Cd-poisoned chickens (without any additional treatment) showed a remarkable decrease (*p* < 0.05), however the BW of the protective agent treated-only groups (with α-LA or CGA or the combination of both) did not change significantly. When compared with the Cd poisoning group, the body mass of each protective agent in isolation and Cd cotreatment group increased by varying degrees, however, without any significant difference. The combined treatment of α-LA and CGA showed no obvious effect in Cd + α-LA + CGA group compared to the treatments in isolation. 

Figure 1 illustrates the effects of α-LA and CGA on the organ coefficients of Cd-poisoned chickens. When the treatments were compared with the blank control group, the liver and heart coefficients in the Cd poisoning group were significantly decreased, while the kidney and brain coefficients were remarkably increased (*p* < 0.05). Slight alternations were observed in α-LA, or CGA and Cd cotreatment groups. All four organ coefficients showed no significant changes in the α-LA + CGA + Cd group when compared with the Cd + α-LA group and Cd + CGA group.

### 3.2. Effects of α-Lipoic Acid and Chlorogenic Acid on Serum Biochemical Markers of Cadmium-Poisoned Chickens

The influence of α-LA and CGA on the indicators of liver protein metabolism in the serum of Cd-poisoned chickens is provided in Figure 2A. When compared with the control group, the levels of TP, ALB and A/G of the Cd group decreased significantly (*p* < 0.01). The GLB content increased with no significant differences (*p* > 0.05). After treatment (with α-LA or CGA or their combination) these indicators varied. 

The changes in the levels of hepatic-specific enzymes are demonstrated in Figure 2B. AST and ALP are well recognized indicators that can reflect liver function. AST activity increased significantly in the Cd group when compared with the control group, however, this change was not evidently seen from liver histology (Figure 3B). Neither α-LA nor CGA showed significant effects on restoring the activity of AST. Meanwhile, ALP activity showed no obvious variations in each group treated with different agents.

Figure 2C,D demonstrates that the levels of CREA, UA and CK increased significantly in the Cd-treated group. α-LA and CGA can alleviate the abnormal increases. The number of RBC and the HGB concentrations were also investigated in this research. Figure 2E showed that Cd caused a decrease in RBC and HGB in chickens. The addition of α-LA or CGA produced no obvious effect. The Cd concentration in serum was highly elevated when compared with the control group, which suggested a successful model. 

This research demonstrated that the administration of Cd (50 mg/L) could lead to abnormal liver functions in chickens and the two protective agents could partly reverse this trend.

### 3.3. Effects of α-Lipoic Acid and Chlorogenic Acid on the Histopathology of Cadmium-Exposed Chicken Livers

The histopathological investigation showed that the hepatocytes were arranged regularly with clear hepatic cords in the blank control group, α-LA group, CGA group and α-LA + CGA group (Figure 3A,C–E), but the addition of Cd caused liver damage (Figure 3B). Clearly, Cd group showed vacuolization (black arrows) and swelling of hepatocytes and disorganized hepatic cords, although these variations were mild. These liver changes were reduced after the poisoned chickens were treated with α-LA or CGA. Figure 3F–H showed no significant pathological changes at portal areas, only mild infiltration of heterophilic granulocytes (yellow arrows) was observed. The combined treatment of α-LA and CGA had the strongest effect.

The ultrastructure of the hepatocytes was photographed using a transmission electron microscope (Figure 4). A relatively intact nucleus and homogeneous cytoplasm were observed in the control, α-LA, CGA and α-LA + CGA groups. Cd treatment caused severe damage to the hepatocytes, including in homogeneous chromatin in the cell nucleus, nuclear condensation, vacuolization of cytoplasm and damage and dissolution of the cell double-layer membrane structure. The histological examination clearly demonstrated the protective effect of α-LA or CGA against Cd cytotoxicity in the liver. 

Figure 5 shows the ultrastructure of mitochondria, which was relatively intact in the control group, α-LA group, CGA group and α-LA + CGA group. The cristae rupture and vacuolization in the mitochondria of the liver tissue were observed after Cd treatment. Figure 5E–H shows that α-LA and CGA had therapeutic effects against Cd-induced mitochondrial damage.

### 3.4. Effects of α-Lipoic Acid and Chlorogenic Acid Antioxidative Properties on Cadmium-Exposed Chicken Liver 

The changes in the levels of lipid peroxidation, oxygen free radicals and trace elements in the liver tissue are provided in Figure 6. Lipid peroxidative products such as MDA and hepatic-enzymatic and hepatic-non-enzymatic antioxidant indexes (including total superoxide dismutase (T-SOD), catalase (CAT), hydrogen peroxide (H_2_O_2_), glutathione S—transferase (GST), glutathione (GSH), glutathione peroxidase (GSH-Px) and total antioxidant capacity (T-AOC)) were measured in our study. The levels of MDA, GSH and H_2_O_2_ were significantly increased in the Cd-treated group when compared with the control group. The activities of T-SOD, GST, GSH-Px, CAT and T-AOC were significantly decreased in the Cd-treated group when compared with the control group. All of these indicators, except CAT, changed by varying degrees after the groups were treated with the protective agents. Table 3 illustrates the trace element contents in the liver of the chickens in this experiment. A sharp decrease in the levels of trace elements such as Mn, Fe, Cu, Zn and Se existed in the Cd-treated chickens when compared with the normal chickens. Treatment with α-LA or CGA with Cd increased the contents of these elements. Treatment combining α-LA and CGA produced the best effects on Cd poisoning. No Cd was observed in the control, α-LA, CGA and α-LA + CGA groups (Figure 6E). Conversely, a drastic increase in Cd was observed after the chickens were treated with Cd. The Cd concentration in the liver decreased significantly after the poisoned chickens were treated with α-LA and/or CGA.

### 3.5. Effects of α-Lipoic Acid and Chlorogenic Acid on the Mitochondrial Apoptosis Pathway in Cadmium-Exposed Chicken Liver Cells 

Figure 7 shows the results of the immunohistochemical analysis of the chicken livers. Bax expression in the protective agent-treated group was consistent with the blank control group. After the chickens were treated with Cd, a sharp increase in the expression of the proapoptotic protein Bax was observed. When compared with the Cd-only group, the levels of Bax in the α-LA or CGA and Cd cotreatment groups significantly decreased (*p* < 0.05) and reduced with a more remarkable significance in the α-LA + CGA + Cd group (*p* < 0.01). The mitochondrial apoptosis pathway is one of the most important apoptosis pathways; therefore, we included this pathway in our investigation. The mRNA expression of Bax, Cyt C, caspase 3 and caspase 9 drastically increased in the Cd-poisoned chickens when compared with the healthy control group (Figure 8). The mRNA expression of Bcl-2 and Bcl-2/Bax was drastically decreased in the Cd group. The inclusion of the two protective agents produced a statistically significant decrease in the expression of Bax, Cyt C, caspase 3 and caspase 9 and increased the levels of Bcl-2 and Bcl-2/Bax (*p* < 0.05 or *p* < 0.01).

## 4. Discussion

Cd, a well-known noxious heavy metal and environmental pollutant worldwide, presents extensively in various heavy industries, plastics, paints, soil, contaminated water, air and food [29]. It is mainly transferred through the food chain, eventually entering the human body and causing serious diseases. Consequently, Cd has aroused increasing interest among scientific researchers. The adverse impact of Cd on chicken growth performance is evident in many aspects such as body mass, organ coefficient, food consumption and water intake. The degree of weight gain and testis coefficient decreased in CdCl_2_-treated male chickens, and different Se sources showed strong antagonistic action against Cd [30]. These results were consistent with our findings. Cd inhibited the growth performance of Cd-exposed chickens, but the combination of the two protective agents, α-LA or CGA, reversed this negative consequence to some extent. However, the organ coefficients of the experimental animals did not show any significant change after treatment with α-LA or CGA alone, or with their combination when compared with the Cd-treatment group.

One important target of Cd is the mitochondria, which is the major source of body energy and endogenous ROS. The content of ROS, along with the level of antioxidants, maintains a balance under physiological conditions. Once unbalanced, a type of damage mechanism called oxidative stress occurs, resulting in cell death and the disordered internal environment [31]. Therefore, antioxidants have become an effective way to counteract Cd toxicity [32]. The present study further confirmed that the administration of two antioxidants, α-LA and CGA, demonstrated protective action against Cd intoxication. As a vital organ for detoxification, the function of the liver reflects the development of Cd poisoning. AST is one of the most often measured parameters for evaluation of liver damage [33], while ALP is a less liver specific marker which also has a good sensitivity for liver disease such as cholestatic disease [34]. Studies have shown different concentrations of Cd (40, 80 and 120 mg/kg diet) caused significant increase in AST and ALP activities, which may be attributed to the hepatotoxic effect [33]. However, our results of α-LA and CGA did not show any effective actions in recovering the activity of AST after Cd treatment and ALP activity did not show any significant changes in each group. This may be related to the dosage, exposure time and injection method of protectors. In addition, some assays of TP, GLB and ALB also manifest the metabolism, biosynthesis and detoxification capability of the liver [35]. Therefore, our research included these indicators. Our findings showed that the content of TP, ALB and the ratio of A/G decreased significantly while GLB content increased slightly after Cd exposure, which is consistent with the research conducted by Oladipo et al. [36]; a decrease in ALB may be due to hepatic impairment and/or increase in urinary excretion, resulting from damaged renal function [36]. In our research, the drop of ALB was more likely to be attributed to renal damage. Simultaneously, the extent of damage in Cd-exposed chickens treated with α-LA or CGA decreased, suggesting their protective effects exerted by stabilizing the cell membrane of the hepatocytes. It is worth noting that compared with Cd and the single protector group, the combined use of α-LA and CGA showed a synergistic effect on restoring liver function.

The kidney plays an irreplaceable role in excreting metabolic waste and regulating systemic homeostasis [37]. UA and CREA are two common indexes in the serum of mammals that are used to evaluate the degree of kidney injury. CK in blood is an indirect biomarker for cardiovascular disease, which was significantly increased in Cd-exposed rabbits [38]. In our results, the contents of UA and CREA and CK activity significantly increased after Cd treatment, but rapidly decreased when α-LA or CGA was administrated to the poisoned chickens. The variation in hematological parameters is considered a sensitive early indicator of trace element toxicity, such as Cd and Pb [39]. Therefore, we also detected the number of RBCs and the concentration of HGB. We observed that Cd treatment significantly decreased HGB concentration, while the combination of protective agents changed this tendency. However, the number of RBCs merely dropped slightly in the Cd group. Hyogo et al. have proved that chronic Cd poisoning can result in the injury of liver and kidney and hemolysis of RBC with a subsequent anemia; although the iron accumulates in organs because of hemolysis, the normal function of erythropoietin (EPO) was suppressed by Cd, which failed to compensate for the hemolytic anemia [40]. The decrease of HGB also indicated the occurrence of anemia.

Histological changes intuitively reflect the degree of injury in various tissues and organs. In this study, the results of the pathological analysis showed that mild liver damage was observed in the Cd-model group, including vacuolization, swelling of hepatocytes and disorganized hepatic cords. This pathological change was not as severe as those studies that add Cd into the basal diet [41], which may be partly due to the method of Cd entering into the body. The ultrastructure of the liver tissue was clearly observed using transmission electron microscopy. The liver tissue exhibited different extents of damage in the nucleus and mitochondria of the hepatocytes, including nuclear condensation, vacuolization of cytoplasm and cristae rupture. These results are consistent with previous research demonstrating that Cd caused obvious ultrastructural changes [12]. The damage to the chicken liver cell structure was alleviated after the administration of α-LA and CGA to intoxicated chickens. This result is related to the chelating mechanism of thiol-containing chelating agents and metal ions, eliminating ROS and exerting a protective effect [42]. However, no significant difference was observed in the combined-treatment group when compared to the α-LA-alone or CGA-alone groups. Nevertheless, different histological changes induced by Cd have also been previously reported. Coagulation necrosis and congested vessels were observed in the liver of Wistar rats exposed to Cd for 15 days [43]. The variations in results may be partly due to different routes of administration, dosage, exposure time and so on.

A large number of studies have shown that Cd leads to an unbalanced redox state by promoting the production of hydroxyl radicals and superoxide radicals. Teng et al. have studied that MDA and Cd content were increased and activities of antioxidant enzymes (SOD and GSH-Px) were significantly reduced after Hailan white chickens were exposed to Cd [44]. MDA is the product of lipid peroxidation and indirectly reflects the degree of free radical attack on cells, which is often measured with SOD activity, an enzyme that represents the capacity of scavenging oxygen free radicals in the body [45]. Other studies have shown that in the serum and liver of chickens, the exposure of Cd significantly reduced the activities of SOD, GSH, GSH-Px and CAT while improving MDA and H_2_O_2_ level [46]. In effect, GSH is a cofactor for GSH-Px activity, which, with CAT, is involved in the removal of H_2_O_2_ [47]. Therefore, some antioxidant and oxidative stress parameters were analyzed in the present study to assess oxidative damage to the liver resulting from Cd. These results supported previous research; these indexes clearly indicated that Cd induced an imbalance between oxidation and antioxidation status [48,49]. The combination of α-LA and CGA is more effective than a single drug treatment. It is well established that Zn and Cu are important components of SOD, whose structure is stabilized by trace elements [50]. In addition, Se is the active site of GSH-Px. Previous studies have illustrated that Cd treatment alone significantly decreased the content of Mn, Fe, Cu, Zn, Se and Cr in chickens, as measured by ICP-MS. Cotreatment with Se increased the concentrations of these elements, which corresponds with the observations in our study. With respect to the detection of poisons in the liver, the load of Cd dropped significantly with the addition of α-LA or CGA. The above results demonstrated that these protective drugs alleviated oxidative injury caused by Cd, and were responsible for the increase in key trace elements. It should be noticed that the metabolism of Zn (a cofactor in proteins involved in oxidative defenses) will be disturbed by Cd toxicity, leading to unstable Cd/Zn ratio [51].

Apoptosis is a type of programmed cell death regulated by specific genes. A number of experiments have demonstrated that Cd could trigger cell apoptosis [11,52,53] by activating the mitochondrial pathway [54]. The activation of caspase 3 occurred because of Cd exposure, while the expression of major genes in the mitochondrial pathway changed, including caspase 9, cyt c, Bcl-2 and Bax [55]. It has been reported that the antiapoptotic protein Bcl-2 controls the release of Cyt C by regulating mitochondrial function [56]. Our results are in accordance with previous data. The immunohistochemical detection revealed that the expression of Bax increased dramatically in the Cd group. The mRNA levels of Cyt C, caspase 3, caspase 9 and Bax/Bcl-2 showed an obvious increase in the liver of Cd-model chickens, suggesting the occurrence of apoptosis. α-LA and CGA reduced the severity of Cd toxicity by relieving the mitochondrial damage induced by oxidative stress and the production of ROS, revealing obvious therapeutic effects on Cd intoxication. It is worth mentioning that the combined addition of these two drugs was more effective in inhibiting the mitochondrial apoptosis pathway. Our results support the hepatoprotective effects of α-LA and CGA on Cd-induced injury in vivo.

## 5. Conclusions

In summary, the current study confirmed that α-LA and CGA are capable of reversing oxidative toxic reactions. They partially or fully alleviated the inhibition of growth (body weight) of Cd-exposed chickens, normalized biochemical indicators, liver function, cell nucleus and mitochondria structure, antioxidative agent activities and inhibited cell apoptosis induced by the mitochondrial-dependent pathway, thus exhibiting protective action. These data suggest that α-LA and CGA are capable of inducing scavenging free radicals and chelating metal ions, to reduce Cd toxicity. Our findings provide new insights into the attenuation of Cd-induced hepatotoxicity.

## Figures and Tables

**Figure 1 animals-11-01606-f001:**
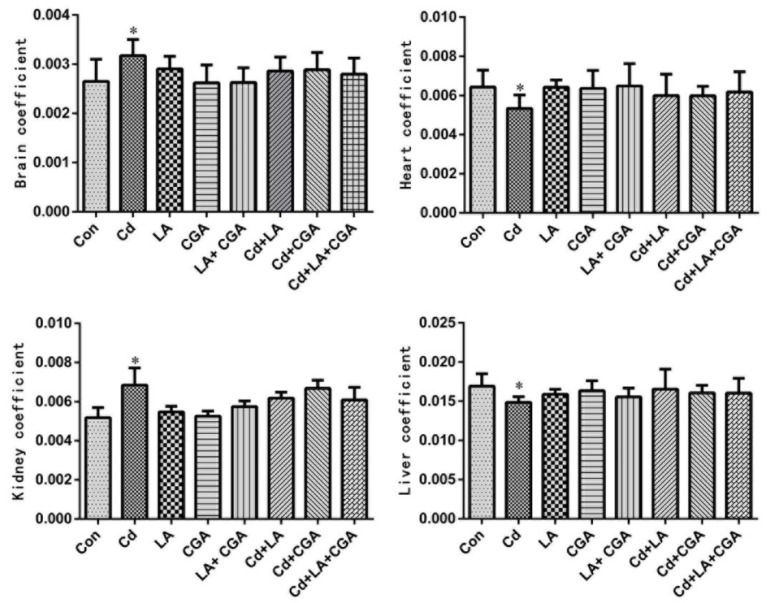
Effects of α- lipoic acid and chlorogenic acid on the coefficients of four different organs of cadmium poisoned chickens (*n* = 3, x ± s). Electronic balance was used to weigh heart, liver, kidney and brain. Organ coefficient % = (organ mass/body mass) × 100%. * *p* < 0.05 vs. control.

**Figure 2 animals-11-01606-f002:**
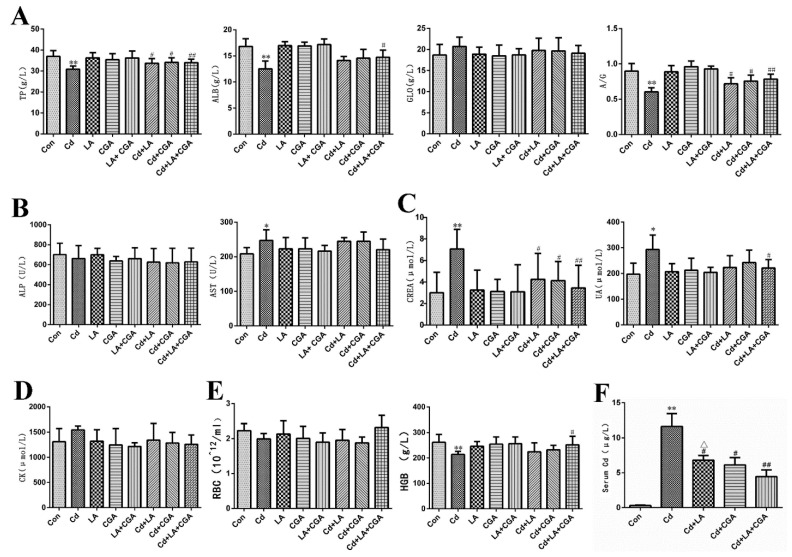
Effects of α-lipoic acid and chlorogenic acid on serum biochemical markers of cadmium-poisoned chickens. (**A**) The content of TP, ALB and GLB and the ratio of ALB/GLB was measured by the automatic blood biochemical analyzer. (**B**) The activities of ALP and AST are specific hepatic-enzymes. The method was same as above. (**C**) The activities of CREA and UA were detected to indicate renal injury. (**D**) The variation of CK represents the cardiac function. (**E**) RBC and HGB are indexes about hematological changes. (**F**) ICP-MS was used in measurement of concentration of Cd in serum. * *p* < 0.05, ** *p* < 0.01 vs. control, ^#^
*p* < 0.05, ^##^
*p* < 0.01 compared to the Cd-treatment group and ^Δ^
*p* < 0.05 compared to the α-LA + CGA + Cd group.

**Figure 3 animals-11-01606-f003:**
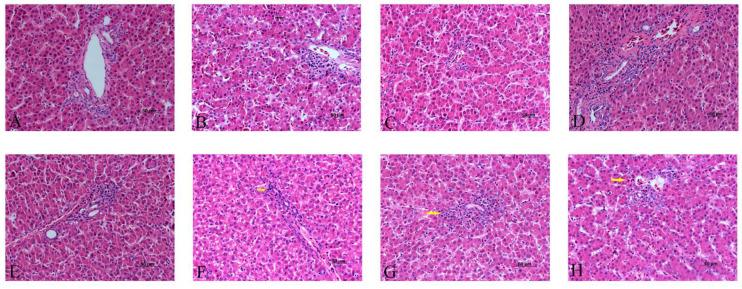
The effect of α-lipoic acid and chlorogenic acid on histopathology of cadmium-exposed chicken liver (50 μm). (**A**): Control group, (**B**): Cd group, (**C**): α-LA group, (**D**): CGA group, (**E**): α-LA + CGA group, (**F**): Cd + α-LA group, (**G**): Cd + CGA group and (**H**): Cd + α-LA + CGA group.

**Figure 4 animals-11-01606-f004:**
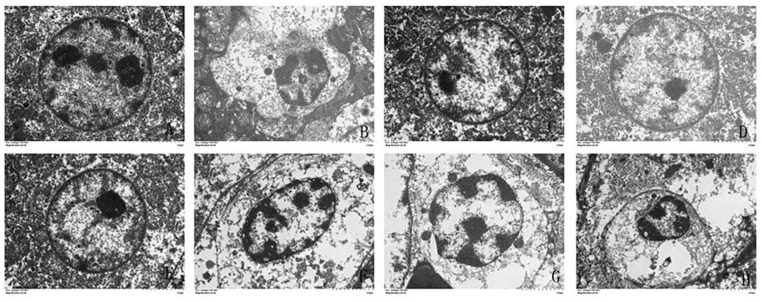
The effect of α-lipoic acid and chlorogenic acid on the nuclei of hepatocytes in cadmium-exposed chickens (6000×). (**A**): Control group, (**B**): Cd group, (**C**): α-LA group, (**D**): CGA group, (**E**): α-LA + CGA group, (**F**): Cd + α-LA group, (**G**): Cd + CGA group and (**H**): Cd + α-LA + CGA group.

**Figure 5 animals-11-01606-f005:**
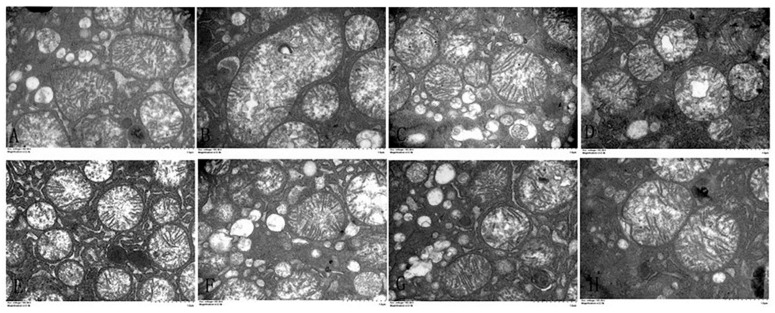
The effect of α-lipoic acid and chlorogenic acid on mitochondria of cadmium-exposed chicken liver (12,000×). (**A**): Control group, (**B**): Cd group, (**C**): α-LA group, (**D**): CGA group, (**E**): α-LA + CGA group, (**F**): Cd + α-LA group, (**G**): Cd + CGA group and (**H**): Cd + α-LA + CGA group.

**Figure 6 animals-11-01606-f006:**
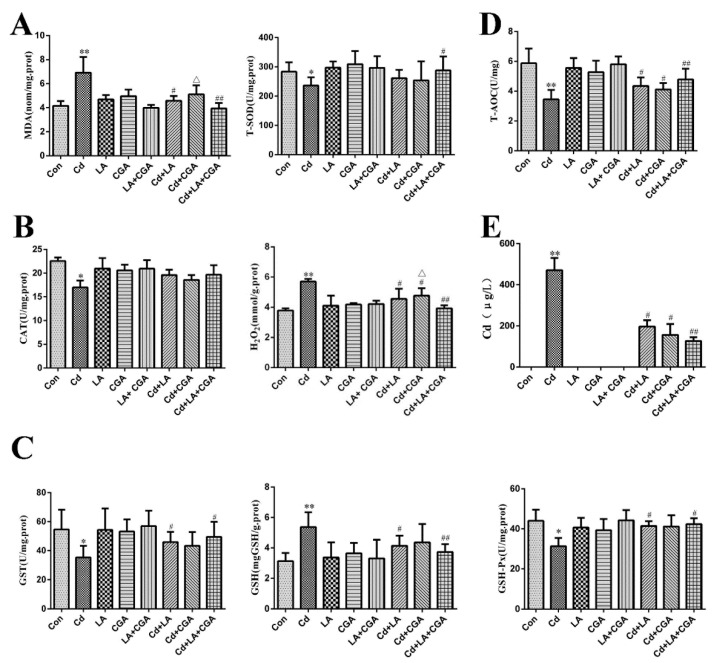
Effects of α-lipoic acid and chlorogenic acid on antioxidative properties and Cd concentration of cadmium-exposed chicken liver. (**A**) Thiobarbituric acid (TBA) method was used to measure the content of MDA. The T-SOD assay kit was used to detect the activity of T-SOD. (**B**) The activities of CAT and H_2_O_2_ was measured by spectrophotometry according to CAT and H_2_O_2_ assay kits. (**C**) Antioxidant parameters—GST, GSH and GSH-Px activities were detected by the GST, GSH and GSH-Px assay kits. (**D**) The activity of T-AOC represents total antioxidant capacity of body. (**E**) Concentration of Cd in liver was detected by ICP-MS. * *p* < 0.05, ** *p* < 0.01 vs. control, ^#^
*p* < 0.05, ^##^
*p* < 0.01 compared to Cd-treatment group and ^Δ^
*p* < 0.05 compared to α-LA + CGA + Cd group.

**Figure 7 animals-11-01606-f007:**
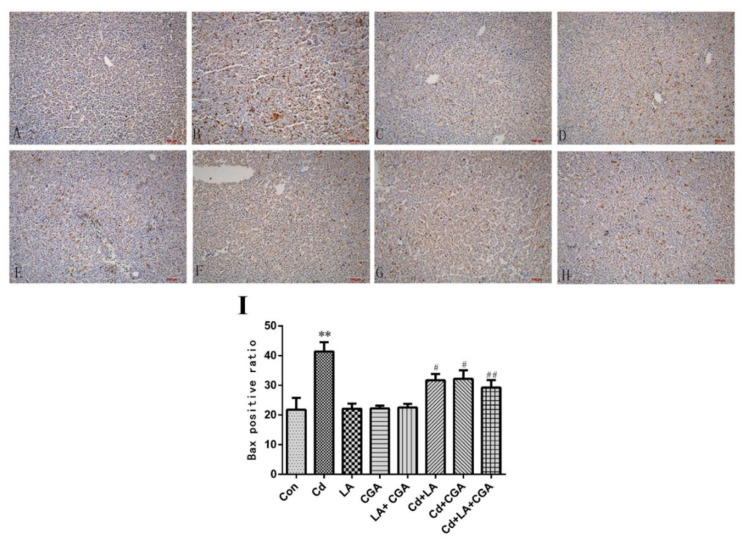
Effects of α-lipoic acid and chlorogenic acid on apoptosis protein Bax in cadmium-exposed chicken liver detected by immunohistochemical (200×). The photographs of immunohistochemistry—(**A**) control group; (**B**) Cd group; (**C**) α-LA group; (**D**) CGA group; (**E**) α-LA + CGA group; (**F**) Cd + α-LA group; (**G**) Cd + CGA group; (**H**) Cd + α-LA + CGA group. (**I**) The positive ratio of Bax in all groups. ** *p* < 0.01 vs. control, ^#^
*p* < 0.05, ^##^
*p* < 0.01 compared to the Cd-treatment group.

**Figure 8 animals-11-01606-f008:**
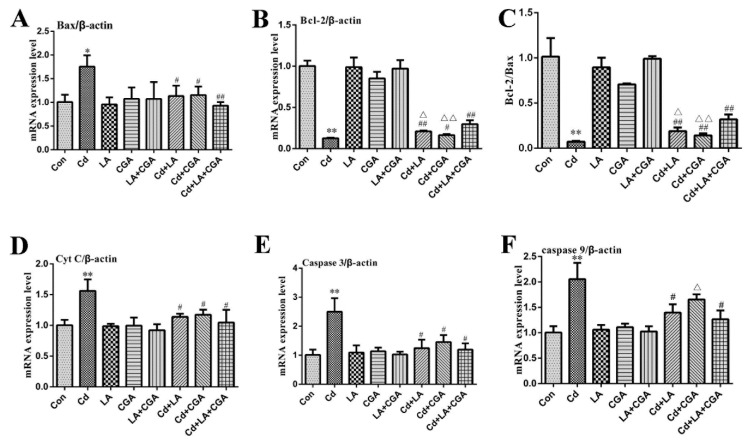
Effects of α-lipoic acid and chlorogenic acid on mitochondrial apoptosis gene in cadmium poisoned chicken liver. The mRNA levels of (**A**) Bax; (**B**) Bcl-2; (**C**) ratio of Bcl-2/Bax; (**D**) Cyt C; (**E**) Caspase 3; (**F**) Caspase 9 were measured by q-RT PCR. * *p* < 0.05, ** *p* < 0.01 vs. control, ^#^
*p* < 0.05, ^##^
*p* < 0.01 compared to the Cd-treatment group, ^Δ^
*p* < 0.05 and ^ΔΔ^
*p* < 0.01 compared to the *α*-LA + CGA + Cd group.

**Table 1 animals-11-01606-t001:** Sequence of primers for real-time RT-PCR amplification.

Gene Name	Primer Sequence
β-actin	F: CCGCTCTA TGAAGGCTACGC
	R: CTCTCGGCTGTGGTGGTGAA
Bax	F: TCCATTCAGGTTCTCTTGACC
	R: GCCAAACATCCAAACACAGA
Bcl-2	F: GAGTTCGGCGGCGTGATGTG
	R: TTCAGGTACTCGGTCATCCAGGTG
Cyt C	F: CTTCTTCCTCCTGGTGAACG
	R: GCACTCCGAAAGTCTCCTGA
Caspase 3	F: CTGAAGGCTCCTGGTTTA
	R: TGCCACT CTGCGATTTAC
Caspase 9	F: ATTCCTTTCCAGGCTCCATC
	R: CACTCACCTTGTCCCTCCAG

**Table 2 animals-11-01606-t002:** The body weight of experimental chickens in each group at the day of 0, 30, 60 and 90.

Group	Initial Weight (g)	30 d Weight (g)	60 d Weight (g)	90 d Weight (g)
Control	76.6 ± 6.6	386.2 ± 59.1	761.5 ± 138.9	1067.0 ± 215.7
Cd	79.2 ± 8.3	327.1 ± 32.9 *	644.5 ± 66.9 *	892.1 ± 66.5 *
α-LA	81.3 ± 6.0	373.6 ± 43.1	784.2 ± 62.7	1107.7 ± 108.3
CGA	78.2 ± 6.2	364.5 ± 37.7	764.1 ± 99.0	1108.0 ± 102.0
α-LA + CGA	80.8 ± 6.4	376.1 ± 40.4	757.0 ± 83.4	1061.2 ± 76.9
Cd + α-LA	76.1 ± 3.1	350.4 ± 30.1	692.0 ± 78.8	994.6 ± 123.2
Cd + CGA	79.5 ± 5.6	360.6 ± 54.8	764.2 ± 86.3 ^#^	995.7 ± 158.1
Cd + α-LA + CGA	77.9 ± 5.2	360.3 ± 31.8	726.2 ± 96.0	993.9 ± 107.7

Note: Results represent mean ± SD of three samples per group. Initial weight, 30 d weight, 60 d weight and 90 d weight were body weight (BW) of 9-, 39-, 69- and 99-day old chickens respectively. The BW of each experimental animal was measured by electronic balance. * *p* < 0.05 vs. control, ^#^
*p* < 0.05 compared to the Cd-exposure group.

**Table 3 animals-11-01606-t003:** Contents of trace elements (Mn, Fe, Cu, Zn and Se) in the liver of experimental chickens.

Group	Mn (μg/g)	Fe (μg/g)	Cu (μg/g)	Zn (μg/g)	Se (μg/g)
Control	16.69 ± 1.78	565.22 ± 17.09	20.61 ± 0.76	175.08 ± 28.23	4.13 ± 0.20
Cd	7.23 ± 0.54 **	354.72 ± 37.35 **	11.62 ± 0.89 **	103.61 ± 3.58 **	2.75 ± 0.12 **
α-LA	17.8 ± 1.37	668.32 ± 138.6	21.76 ± 2.92	181.62 ± 7.18	4.32 ± 0.75
CGA	17.71 ± 0.68	606.53 ± 119.80	23.50 ± 1.00	184.24 ± 7.84	4.57 ± 0.15
α-LA + CGA	16.96 ± 1.57	582.98 ± 43.73	22.85 ± 2.04	173.39 ± 5.95	4.34 ± 0.18
Cd + α-LA	11.59 ± 0.99 ^#^	468.42 ± 35.83	15.48 ± 0.54 ^#^	130.46 ± 6.76 ^#^	3.39 ± 0.61
Cd + CGA	9.74 ± 0.57 ^#^	418.42 ± 35.83	14.03 ± 1.48	115.22 ± 6.61 ^Δ^	2.98 ± 0.32
Cd + α-LA + CGA	12.43 ± 0.79 ^##^	477.46 ± 17.78 ^#^	17.24 ± 1.17 ^#^	154.32 ± 5.77 ^#^	3.82 ± 0.25 ^#^

Note: Results represent mean ± SD of three samples per group. Concentrations of trace elements were detected by ICP-MS. ** *p* < 0.01 vs. control, ^#^
*p* < 0.05, ^##^
*p* < 0.01 compared to the Cd-exposure group and ^Δ^
*p* < 0.05 compared to the α-LA + CGA + Cd group.

## Data Availability

Data are available from the corresponding author upon reasonable request.

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
