# Peer review of "Protective Effects of α-Lipoic Acid and Chlorogenic Acid on Cadmium-Induced Liver Injury in Three-Yellow Chickens"

_animals, 2021, doi:10.3390/ani11061606_

Round 1
Reviewer 1 Report
This is a well written and presented paper. I am happy to accept it after minor corrections.
Line 64-66 provide a reference.
M&Ms section give references for the methods used and specifications for all instruments
Line 82-83 give more reasons "why" you do this research and better put it in the end of the introduction
line 104-116 present the importance of CGA in research
Line 308 write the scientific names of the enzymes listed.
Reviewer 2 Report
The work is focused on the extremely important topic of searching for natural protectors reducing the hepatotoxicity of cadmium. Such studies are often carried out in vitro, meanwhile the presented study concerns an prolonged experiment with birds aged 1 day to 3 months. Cadmium is one of the most dangerous food toxins, it exhibits multi-organ toxicity ranging from lung and kidney cancer to hypertension and osteoporosis. The target organ of cadmium action remains the liver which is responsible for its detoxification. It is also a common environmental toxin, hence the discussed topic is of great importance also for the protection of human health. I am impressed by the enormous number of analyzes used at work to assess cadmium hepatotoxicity: histological evaluation, biochemical and haematological indicators as well as mitochondrial pathway-induced apoptosis factors. They also studied the effect of cadmium on the state of mineral balance in the body, The results clearly show that both the α-LA and CGA can be beneficially used in alleviating oxidative stress and limits Cd-triggered hepatotoxicity.
In general experiment was properly arranged and described. It is clearly written and technically sound. The depth in the research study and its relevance with the research methodology as well as graphical presentation of obtained results are suitable for publication in Animals. However, before publication some corrections and completions are required.
Crucial Comments
Title:I suggest to add breed of chickens
Abstract:
Line 30: modify “1-day-old chicks of broiler Three-Yellow breed”
Introduction:
Line 70: antioxidant activity?
Line 78: add reference e.g. DOI: 10.1016/j.jtemb.2018.06.015
Line 82: Explain why did you used chickens (as model organism or due to their exposure to cadmium during breeding?)
Line 89: add reference e.g. DOI: 10.1080/10934529.2012.672133
Line 121: add reference e.g. DOI: 10.1016/j.fct.2019.110751
Line 122: Add the source of Animals exposure to cadmium
Material and Methods:
Line 135-136: explain how the doses of Cd and both protectors were established (if based on literature, cite it)
Line 136: add the details about dosing frequency e.g. each day
Line 139: add the code of Ethics Commission approvement
Line 141: to complete description of ICP-MS procedure (mineralization process)
Line 153: add the device name and producer
Line 165: add the microscope used and magnification
Line 174: add the used microscope type
Line 181: the kits should be specified
Line 187: Needs completing, the ICP-MS equippement and analitycal details are lacking
Results:
Authors should specify the type of exposure to cadmium, be it acute or chronic. If they measured the volume of drinking water, they can determine the dose taken per week (it takes into account the accumulation of cadmium in the tissues) and throughout the experiment. It is extremely important in toxicological studies.
Line 293: Add the description of each Photo as legend, if it is the same as for Fig. 3 mark it
Line 301: the same as above
Line 322: Fig.5 add the description to each chart
Line 347: Explain the letters A-H on Photos or delete thea if unnecessary
Line 352: Explain each chart as legend
Line 369: The antagonism Cd-Zn should be noticed
Line 385: It should be emphasized the effect of both protectors combination, was the synergism observed or not?
Line 404: Add work focused on birds e.g. DOI: 10.1016/j.jtemb.2018.06.015
Line 440: quote the said paper
Reviewer 3 Report
The article “Protective effects of α-lipoic acid and chlorogenic acid on cadmium-induced liver injury in chickens” by Shi et al involves chicken feeding trials. However, no information regarding ethical approval for the use of animals in for research is provided. The introduction also needs to be reorganised and improved. Besides this, there are additional comments that the authors must address.
Line 23-24: Please clarify and rephrase the sentence “However, the addition of α-LA or CGA significantly mitigated the above variations” and replace the word “on” with the word “in” between the words “effect” and “attenuating”.
Line 28: Please replace the words “unbalanced oxidative status” with oxidative stress”.
Lines 30-39: Please provide the results briefly in abstract and reorganize by removing methodology details.
Lines 48-51: Please replace these lines with information relevant to chicken.
Lines 70-74: Please clarify the statements “Studies have shown that oxidative stress is one of the mechanisms responsible for Cd toxicity, inducing a change in antioxidase activity, producing lipid peroxide, and dis-rupting intracellular sulfhydryl homeostasis [9]. The absence of Nrf2 (an important tran-scription factor involved in regulating oxidative stress in cells) causes mice to become more sensitive to Cd exposure due to an attenuated antioxidant response [10].” and provide a link with this study.
Lines 92-93: Please remove these lines and provide information regarding the effects of α-lipoic acid and chlorogenic acid on cadmium toxicity in chicken.
Lines 95-96: Please provide reference.
Lines 99-100: Please clarify the statement “Oxidized ascorbate, GSH, coenzyme Q, and vitamin E were re-produced after α-LA was administered to Cd-treated male rats [19].” And rephrase to improve the understanding.
Lines 106-108: Please mention the connection between CGA chelation with iron and the formation of hydroxyl radicals.
Lines 110-115: Please remove these lines and provide information relevant to instant study.
Lines 414-424: Please remove these lines and add information relevant to the present study.
Line 454: Please mention which growth parameters were improved and what protocol was used for measuring those parameters. The material and method section does not contain any information regarding the measurement of growth performance.
Reviewer 4 Report
Review of “Protective effects of α-lipoic acid and chlorogenic acid on cadmium-induced liver injury in chickens”
Overall, this paper is a thorough investigation of the effects of cadmium induced liver injury as well as potential amelioration of these effects by α-lipoic acid and chlorogenic acid. The writing is overall strong with minimal corrections needed. However, there are problems with the interpretation of the clinical pathology and documentation of the histological lesions.
Clinical Pathology
- Lines 252-253: “The activities of these two enzymes are inhibited by Cd and restored to some extent by α-LA or CGA.” Your data as reported in Figure 2 does NOT support this statement. ALP is not significantly changed in any of the groups compared to normal. This finding is not surprising; ALP is a measurement of bile stasis which neither you nor any of your cited papers cite as a finding in Cd toxicity. AST is a leakage protein that increases with hepatocellular necrosis. So this statement needs to be changed with the explanation (I just gave) of why these results are consistent with the rest of the data. Any veterinary clinical pathology book that includes avian species should work as a reference.
- You do not sufficiently address the findings of renal toxicity. An increase in UA and creatinine as high as your reported in the absence of dehydration (as indicated by the normal red blood cell concentration) is indicative of significant renal damage. Ideally, renal histopathology would have been performed as well, but if this was not done a suggestion should be made to follow up this finding in future projects.
- Creatinine kinase (CK) is a non-specific indicator of either myocardial degeneration OR skeletal muscle degeneration.
- Globulin is produced by both immune cells and the liver; albumin is liver specific, but may be lost through the kidney due to renal damage. So a drop in albumin in the absence of a drop in globulin could represent either liver or renal damage. The liver pathology does not appear severe enough to have resulted in a drop in albumin production (typically most of the liver would have to be lost for this to occur), so in this case, I suspect renal loss. Further explanation of this result is needed in the paper.
- Address the finding of normacytic, homochromic CBC findings (usually this is attributed to iron deficiency – perhaps Cd is interfering with iron chelation?).
Histopathology
- If possible, include renal pathology (see above).
- The panel for the liver is too small to see any of the lesions you indicate (even magnified on the computer). Further, the photos are taken at different location (A= mid zonal; B, E and G = portal; C,D,F and G = central) hampering direct comparisons. I would suggest a 2x4 panel with much larger photos or even multiple photos for each group (1 portal and 1 central).
- The inflammation indicated while difficult to make out looks like it may be extramedullary hematopoiesis. I would highly recommend a board certified veterinary pathologist with expertise in avian pathology review these results.
- Other than small language usage errors which can be corrected in the final editing, please define all abbreviations at the time of first usage.
The relevance and interest of this topic are high so I would encourage the authors to make the appropriate changes and resubmit.
Round 2
Reviewer 3 Report
The manuscript requires language check and sentence structure. Afterwards, it may considered for publication.
Reviewer 4 Report
Review of revised “Protective effects of α-lipoic acid and chlorogenic acid on cadmium-induced liver injury in chickens”
Many of the main problems with this paper were corrected in the revision, however there are still a few outstanding issues, particularly regarding the histology section. Because I believe this paper to be otherwise strong, I strongly encourage you to make the changes to the histology section.
Section 3.2 (Clin path)
Providing a concise written summary of a large amount of data is always challenging and I liked your grouping the results by organ system.
Line 263-4: While AST is elevated, this is only a very small amount. This detail is important when looking at the liver histology (see below) and should be highlighted here.
Line 266 and 422: CK is a muscle leakage enzyme that can results from skeletal muscle OR cardiac muscle damage (or both). There are myocardial specific CK tests – if one was used that should be mentioned in the M+M
Lines 264 and 406: treatment “failed to restore ALP” but there were no changes in this enzyme even in the cadmium only group.
Section 3.3 (histology)
The interpretation of the liver sections while improved is still problematic. If I blow up the images on a large monitor, I can maybe make out a very minimal increase in microvescicular vacuolization (the specific type shown in the photo) in the cadmium only group. However, the change is within normal variation so it is really hard to attribute to treatment. Similarly, the infiltrates (properly termed either granulocytes OR heterophils) within / around portal areas are not pathologic – just leave those out entirely or just note all sections (including the control photo) include minimal cellular infiltrates considered background. And I do not see any biliary changes at all or evidence of disorganization.
That said, I can see in panel B what appear to be shrunken / degenerating cells that I see rarely in panels C-H and not at all in A. (The black arrow in B which cannot be seen except on magnification is basically sitting on one of these foci). These are the cells which are smaller, darker (hypereosinophilic) or fragmented (look along the left side of the photo) and result in an irregular heaptic cord profile (perhaps what was interpreted as disorganization). These changes are is consistent with the very mild AST elevation. However, a higher magnification photomicrograph would be needed to confirm.
Unfortunately, any other histologic lesions (as present) are minimal. The statement in Line 438 that there was “severe liver damage” is not proved by with the clin path or the histology rpesented. (As you guessed in your reply, I am a specialist in avian pathology and also work as a toxicologist pathologist. I showed the photos (without any other information) to a toxicologist pathologist with 20 years experience in the US National Toxicology Program and she was less convinced than me). Your options are to redo this section with better, more convincing evidence; state that lesions were minimal and consisted of minimal / mild single cell degeneration and necrosis consistent with the clin path and other results; or remove the histology section.
Minor issues (grammar and typos)
Lines 18-22: Unclear wording and use of undefined abbreviations. Suggest: “We evaluated the protective effects of α-Lipoic acid (α-LA) or chlorogenic acid (CGA)and their combination on counteracting cadmium toxicity in vivo in three-yellow chickens. Administration of cadmium (50 mg/L) alone lowered the production performance and resulted in biochemical, histologic and enzyme changes within the liver consistent with hepatic injury induced by oxidative stress and apoptosis of hepatocytes. However, the above variations of Cd group were partially or fully reversed by administration of either α-LA or CGA; their combination showed an even better effect in attenuating Cd-induced hepatotoxicity.”
Abstract: Consider including the same second sentence as the simple summary to orient the reader “We evaluated the protective effects of α-LA, CGA and their combination on counteracting cadmium toxicity in vivo in chickens.”
Line 35: to be more precise change “morphologic changes” to “ultrastuctural changes” to indicate these were electron microscope changes
Line 46: define abbreviation for cadmium (treat summary, abstract and full manuscript as independent publications that can be read and understood on their own)
Line 109: As a reader with a latin / western background, the inclusion of “Chinese herbal medicine” in a discussion of purified biopharmaceutical products confused me. Are these compounds found in some traditional treatments? More common in traditional Chinese pants than say those from the Caribbean? I like the point – just need a bit more context.
Line 167: need a definitive article “The DMI3000B”
Line 199: The way this sentence is written, it implies the same slides stained with H+E were then bleached and used for immunostaining. Is this correct? Or were new slides cut (and not stained with H+E) for the immune which would be more typical.
Line 261: “Hepatic specific enzymes” not hepato
Line 404: you listed ALP twice
